# Dataset of Indicators for the Assessment of Ecosystem Services Affected by Agricultural Soil Management

**Carsten Paul** [1,*], **Cenk Donmez** [1,2], **Petra Koeppe** [1], **James S. Robinson** [1] **and Sonja Barnickel** [1]

1    Leibniz Centre of Agricultural Landscape Research, 15374 Muencheberg, Germany; cenk.doenmez@zalf.de (C.D.); petra.koeppe@zalf.de (P.K.); james.s.robinson9@gmail.com (J.S.R.); sonjabarnickel@yahoo.de (S.B.)
2    Remote Sensing and GIS Laboratory, Landscape Architecture Department, Cukurova University, Adana 01330, Turkey
*    Correspondence: carsten.paul@zalf.de

**Abstract:** Ecosystem services represent an important concept for assessing the sustainability of agricultural management. However, in practical applications, it can be difficult to find indicators suitable for specific services or specific spatial scales. In order to create a toolbox of indicators for assessing the actual or potential supply of ecosystem services in the context of agricultural land and soil management, we conducted a keyword-based literature review in Web of Science Core Collection and SCOPUS, using the terms *ecosystem service* AND *indicator* AND *agricultur\**. The search was performed in January 2019 and was restricted to journal articles written in English. After eliminating duplicates, we identified 180 articles, out of which 121 met our selection criteria. We extracted information on addressed ecosystem services and indicators which used a full-text review. Where studies used ecosystem service definitions other than the Common International Classification of Ecosystem Services (CICES V.5.1), indicators were assigned to the corresponding CICES class or classes. We used the information derived from the review to create factsheets for 37 ecosystem services. Each factsheet provides tables with available indicators applicable at multiple spatial scales that range from field to global, information on the type of input data required, and a reference to the article or articles that the indicator was taken from. The dataset provides a toolbox for researchers to find indicators that fit their respective research needs.

**Dataset:** The MS Word and PDF versions of the dataset are available through the BonaRes Repository (Leibniz Centre for Agricultural Landscape Research (ZALF), Germany, with the following data identification number: https://doi.org/10.20387/bonares-mpzr-ja21 (email required) (accessed on 20 July 2022).

**Dataset License:** The dataset is available under the CC-BY license.

**Keywords:** ecosystem services; indicators; CICES; agriculture; soil management; impact assessment; literature review

## 1. Summary

Ecosystem services represent an important concept for assessing the sustainability of agricultural management. However, in practical applications, it can be difficult to find indicators suitable for specific services or applicable at specific spatial scales. In order to create a toolbox of indicators for assessing the actual or potential supply of ecosystem services in the context of agricultural land and soil management, we conducted a keyword-based literature review in Web of Science Core Collection and SCOPUS. We searched for the terms *ecosystem service* in the title and *indicator* AND *agricultur\** in the title, abstract, or keywords. The search was performed in January 2019 and was restricted to journal articles written in English. Articles were selected if they addressed agricultural land use

and specified indicators for assessing the supply of ecosystem services. After eliminating duplicates, 180 studies were identified by the keyword search, out of which 121 met our selection criteria. Where ecosystem services were based on classifications other than the Common International Classification of Ecosystem Services (CICES V.5.1), indicators were mapped to the matching CICES class or classes. For each indicator, we recorded the recommended and categorized spatial scale based on the type of input data. We recorded all indicators, as presented in the reviewed articles, without filtering or rating their suitability. Missing indicator units were added where they could be inferred from the context of the paper. All corresponding authors were contacted by mail and asked if they agreed with the way their data were recorded. Where authors objected, the records were adapted accordingly. We created a factsheet containing tables of indicators applicable at spatial scales ranging from local to global for each ecosystem service. For each indicator at a specific scale, we provide the category of input data, as well as a reference to the publication(s) that it was taken from.

Since our review is based on a systematic sample of publications, the number of studies addressing each ecosystem service indicates the current research focus and research gaps concerning the supply of ecosystem services affected by agricultural land and soil management. The derived dataset provides a toolbox where researchers analyzing the actual or potential supply of ecosystem services can find indicators that fit their respective research needs.

## 2. Data Description

The dataset contains factsheets for 37 ecosystem services classified in the Common International Classification of Ecosystem Services (CICES) V. 5.1. Information on the structure of CICES and the individual ecosystem services can be found in [1,2]. Our dataset is provided both in MS Word and PDF format to facilitate the easy customization of tables, while also safeguarding against errors in the display of special characters and formulas due to users working with different word-processing software. The selected ecosystem services are relevant in the context of agricultural land and soil management. The factsheets display the full CICES name for each service. They also display a shortened name taken from [3] that can be used, for example, in stakeholder interactions. For each ecosystem service, the factsheets provide tables of indicators for measuring the supply of the services at different spatial scales, ranging from the field scale to the global scale. The indicator tables are based on a systematic review of 180 journal articles published until January 2019. The tables list the indicators' units, what category of input data is used, and which indicators are described by scientific publications. For these publications, we provide complete bibliographic information to reference the use of the indicators in the context of applied research. For many ecosystem services, we did not find indicators for all spatial scales. In these cases, the corresponding factsheets only contain tables for the scales where data were available.

While the total number of scientific publications addressing ecosystem services is too high to facilitate a review of all relevant articles, our keyword-based search generated a systematic sample of studies. Therefore, the number of publications in our dataset for each ecosystem service reflects how often that service is addressed in current research, indicating research focus and possible research gaps (Figure 1).

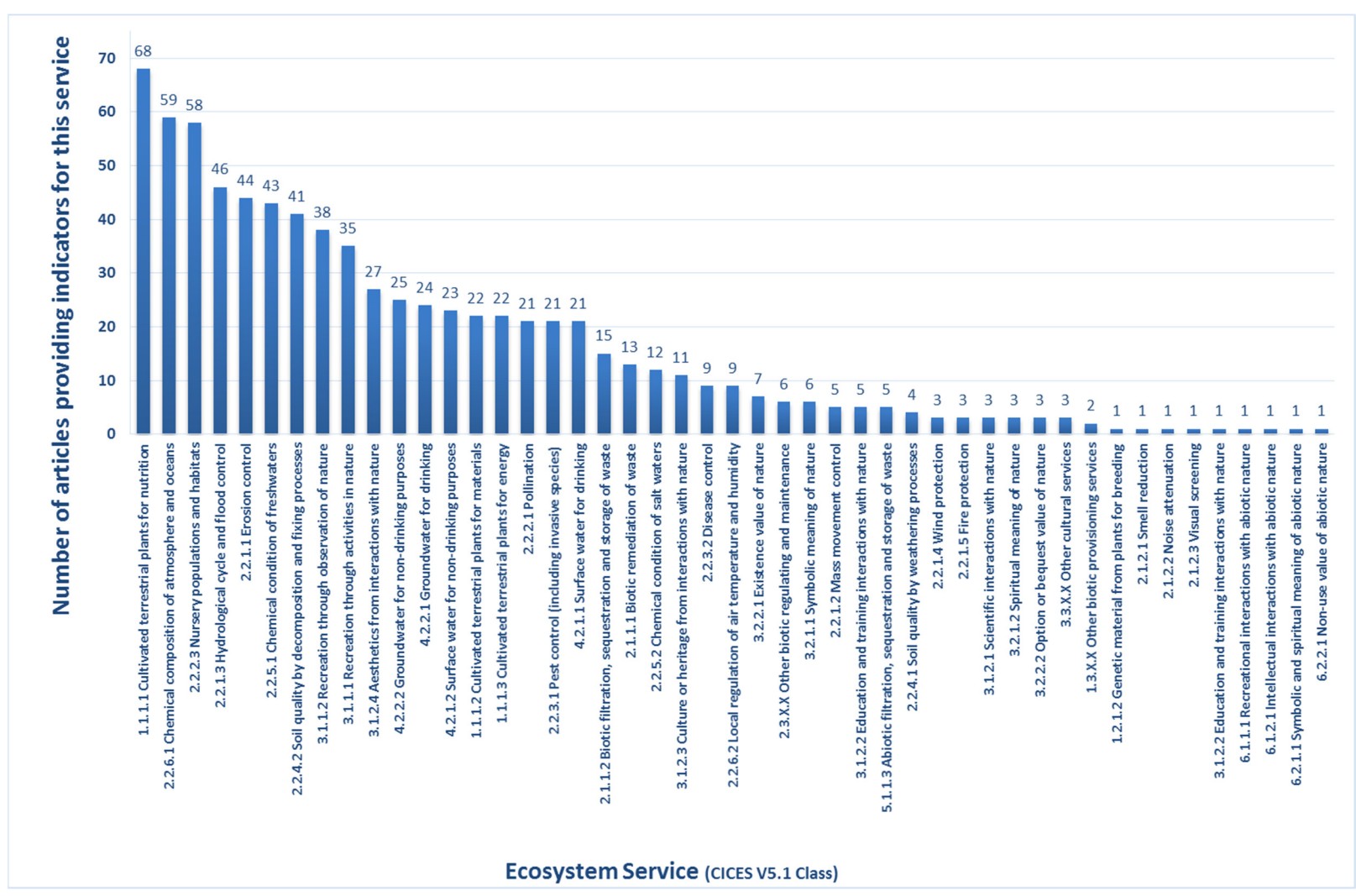

**Figure 1.** Number of studies providing indicators for the ecosystem services contained in our dataset.

The variety of indicators available for each service indicates how far current approaches to measuring its supply differ. Most of all, though, the dataset provides a toolbox for researchers, offering a wide range of indicators for measuring the actual or potential supply of ecosystem services in the context of agricultural land and soil management.

## 3. Experimental Design, Materials, and Methods

### 3.1. Literature Review

The number of publications addressing ecosystem services is too high to allow for a manual review of all articles. At the time of writing, the database Web of Science (WoS) (https://apps.webofknowledge.com/, accessed on 20 July 2022) lists more than 37,000 publications using this term in title, abstract, or keywords. Our goal was, therefore, to extract a systematic sample of articles from which to derive lists of indicators for measuring the supply of ecosystem services in the context of agricultural land and soil management.

We conducted a systematic, keyword-based search in the Core Collection of WoS and SCOPUS (https://www.scopus.com/, accessed on 20 July 2022). On 18 January 2019, we searched for journal articles in English that use the term *ecosystem service* in the *title*, AND *indicator* AND *agricultur\** in *title, abstract,* or *keywords*. In the case of WoS, keywords include both authors' keywords and KeyWords Plus®, which are automatically assigned by the database. The term *ecosystem service* was required to be part of the title to reduce the sample size and limit the publications with a clear focus on this topic. We tested the use of Google Scholar as a third database but rejected it due to the average lower quality of search results, despite more than 10,000 entries being found.

The keyword-based search identified 180 articles. We manually assessed the abstracts and, where necessary, the full text to select those articles that address agricultural land and soil use and provide indicators for assessing ecosystem service supply. In the context of our study, we understand agricultural land and soil use as encompassing all forms of agricultural management with the explicit exclusion of animal husbandry. Consequently, ecosystem services derived directly from agricultural animals, such as the provision of food from animals, such as milk, meat, or honey, were not considered in our study. However, we considered ecosystem services derived from agricultural land use linked to animal husbandry, such as pastures or cropland, including the provision of animal feed. Out of 180 studies identified by the keyword search, 121 met our selection criteria. A manual full-text review of the content was carried out to identify what ecosystem services were addressed, what indicators were used to quantify the supply of the services, and at what scale each indicator was used or what scale it was recommended for. Additionally, we recorded the type of input data used or recommended for the indicators. In this, we distinguished between seven categories, namely experiment or direct measurement, model or GIS, survey, stakeholder participation, expert assessment, statistical or census data, and literature values. This was carried out to facilitate the use of the dataset by looking for indicators which support their research, because information on data requirements makes it possible to consider constraints in data availability, time, or funding during the selection process. For example, indicators which rely on statistical or census data may not work in data-scarce regions. Under such circumstances, indicators based on field measurements or expert assessment may provide an alternative. Likewise, if time or funding are limited, experiments (and indicators based on them) may not be feasible. In this case, indicators based on literature reviews or existing statistics could be selected.

### 3.2. Ecosystem Service Classification

Where ecosystem services were based on a classification other than the Common International Classification of Ecosystem Services (CICES), the indicators were mapped to the corresponding ecosystem service class or classes in CICES, using expert judgement. This could result in one indicator being mapped to multiple ecosystem services and vice versa.

### 3.2.1. Mapping Indicators for Multiple Ecosystem Services to One Ecosystem Service (n:1)

Where studies used ecosystem service definitions more specific than the CICES classification, indicators for multiple services were mapped to the corresponding CICES class that encompasses the specific definitions. For example, if an article provided indicators for "control of wind erosion" and "control of water erosion", they were all mapped to CICES class 2.2.1.1 "Control of erosion rates".

### 3.2.2. Mapping One Indicator to Multiple Ecosystem Services (1:n)

CICES is a strictly hierarchical classification where the final ecosystem service is linked to a specific human use. For example, cultivated crops can provide ecosystem services related to the provision of food (CICES class code 1.1.1.1), the provision of materials (CICES class code 1.1.1.2), or the provision of energy (CICES class code 1.1.1.3). If the end-use is unknown, services can be recorded on a higher hierarchical level of *groups*: "Cultivated terrestrial plants for nutrition, materials or energy; CICES code 1.1.1.x)" [3].

Where we encountered indicators relating to the group level, such as *crop yield [t/ha\*yr$^{-1}$]*, they were assigned to all underlying ecosystem services classes. Figure 2 shows ecosystem services typically addressed together (at group level) in the reviewed literature.

While describing ecosystem services at a higher hierarchical order and as a planned feature of CICES, we also identified a large cluster of services related to water quality and nutrient cycling where ecosystem services overlap. On the one hand, this is due to the special role of soils where biotic and abiotic functions are closely interlinked. On the other hand, it can be explained by specific cases of organic fertilizers such as manure, which can be seen both as a waste product and a source of nutrients. Figure 3 displays the cluster ecosystem services that are closely interlinked in the context of agricultural nutrient management.

When assigning indicators to the corresponding CICES classes, we assessed the context of the reviewed studies and tried to follow the original authors' intention as much as possible. For example, we assessed whether services related to the chemical quality of water referred only to freshwaters or salt waters or whether the authors were exclusively referring to the biotic parts of nutrient cycling or those including the abiotic component. Where the original authors' intent could not be ascertained, we mapped the indicator to all the relevant services.

### 3.3. Data Quality Control

We recorded all indicators, as presented in the reviewed articles, without filtering or rating their suitability. Where information pertaining to an indicator was missing but could be inferred from the context of the article, such as the unit of the indicator, we added the information to the tables. Where the information could not be inferred, we wrote "not provided" in the respective fields of the indicator tables. All corresponding authors were contacted by mail and asked if they agreed with the way their data were recorded. Where authors objected, the records were adapted accordingly.

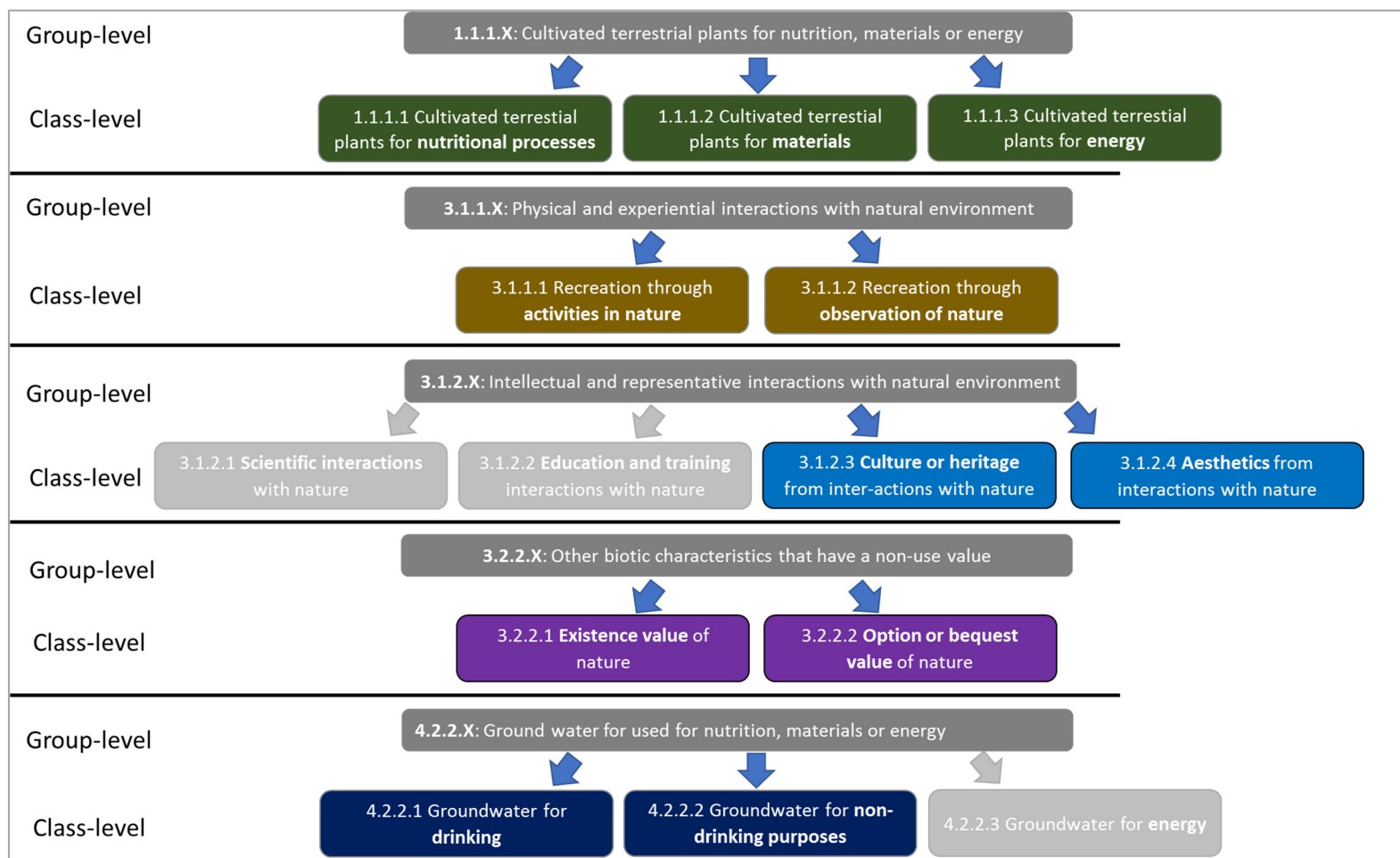

**Figure 2.** Ecosystem services typically addressed together in the reviewed studies are marked in color. Where this occurred, indicators were mapped to multiple ecosystem services.

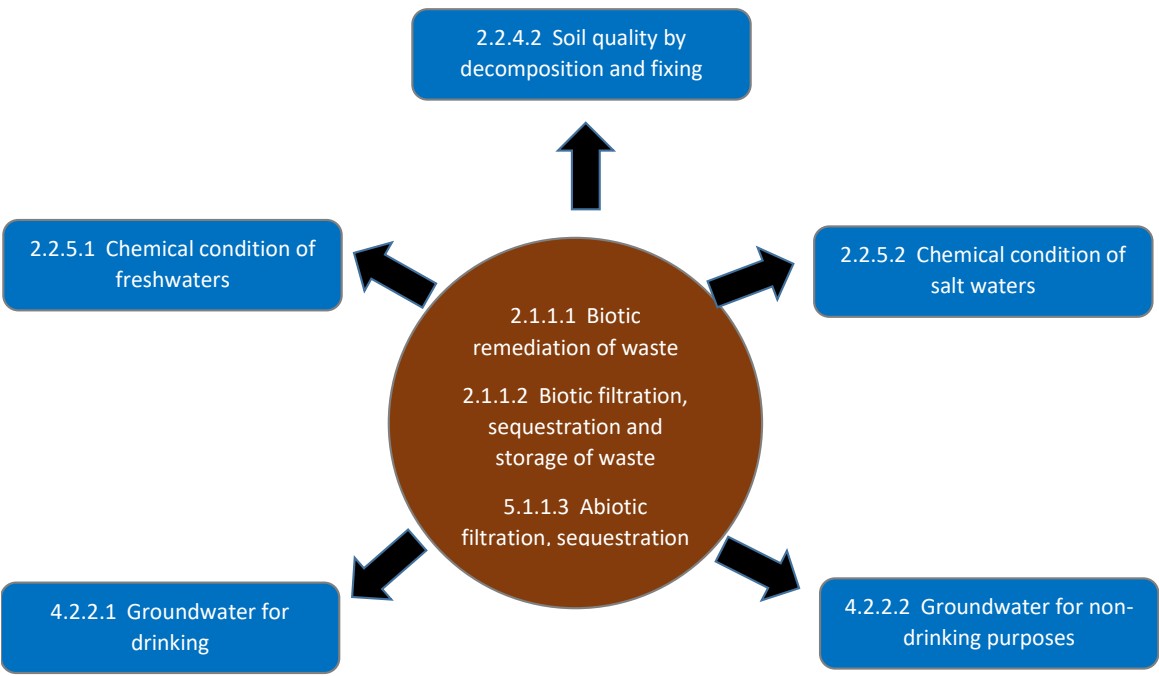

**Figure 3.** Cluster of ecosystem services that are closely interlinked in the context of agricultural nutrient management.

## 4. User Notes

The dataset is provided in MS Word format for the easy customization of tables. However, we are aware that the use of different versions of MS Word or other word-processing software may result in a corrupted display of tables, special characters, or formulas. Therefore, we also included the dataset as a PDF file for reference.

As described in Section 3.2.2 above, some of the ecosystem services are closely interlinked or clustered. When assigning indicators to ecosystem services in the CICES classification, we tried to follow the original authors' intention, as determined by the context of the reviewed studies. However, indicators for closely related ecosystem services may be applicable, even if the study which provided the indictor does not make this connection. We therefore encourage users of this dataset who are searching for indicators in the ecosystem services displayed in Figures 2 and 3 to also check the indicators provided for the related services.

**Author Contributions:** C.P.: conceptualization, methodology, data curation, validation, writing—original draft preparation, supervision. C.D.: data curation, validation, writing—reviewing and editing. P.K.: data curation. J.S.R.: data curation, S.B.: data curation. All authors have read and agreed to the published version of the manuscript.

**Funding:** The study was funded by the German Federal Ministry of Education and Research (BMBF) under the grant scheme BonaRes—Soil as a Sustainable Resource for the Bioeconomy (grant number: 031B 0511B).

**Institutional Review Board Statement:** Not applicable.

**Informed Consent Statement:** Not applicable.

**Data Availability Statement:** The MS Word and PDF versions of the dataset are available through the BonaRes Repository (Leibniz Centre for Agricultural Landscape Research (ZALF), Germany, with the following data identification number: https://doi.org/10.20387/bonares-mpzr-ja21, accessed on 20 July 2022. Download is not restricted and usage is regulated by the CC BY 2.0 (https://creativecommons.org/licenses/by/2.0/, accessed on 20 July 2022) license.

**Acknowledgments:** We would like to thank S. Weigl and S. Nowroz for their invaluable help with editing the factsheets and contacting the authors of the reviewed studies.

**Conflicts of Interest:** The authors declare that they have no known competing financial interest or personal relationships that could have influenced the work reported in this paper.

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
