# Peer review of "Dataset of Indicators for the Assessment of Ecosystem Services Affected by Agricultural Soil Management"

_data, 2022_

Round 1

Reviewer 1 Report

The proposed dataset is well structured and addresses a topic that is certainly interesting, reviewing and organising a wide literature on the topic of ecosystem services indicators. The description of the criteria for the construction of the dataset could certainly be better detailed, for example motivating the choice of the reference period. References could also be improved.

Author Response

Dear reviewer,

In the attached file please find our response to your comments and suggestions.

Reviewer 2 Report

The authors of the data set and descriptor “Dataset of indicators for the assessment of soil-management related ecosystem services” present data about indicators for ecosystem services assessments related to agriculture. (I doubt that only soil-management related ecosystem services where identified since “soil” was not used as search term but “agricultur*”.) Using symbols for the methods that have been used per study is a nice idea but the overall layout of the dataset can be improved. And even though I understand that data descriptors follow a different manuscript style than original research papers (e.g. that no results are presented), the given information is not sufficient.

  • Information about data use is unclear. (The meaning behind this exercise is vague.)
  • The justification for only using SCOPUS and WoS is also not sufficient.
  • Why the term “ecosystem service” was only searched in the title? In addition, in a detailed literature review, the authors would look for synonyms, too. For example, “environ* service*”, “ecol* service*”. Similarly, for “agricultur*” (e.g. farm*). Furthermore, the search terms for only “soil-management” related ecosystem services are not well selected. All agriculture related ecosystem services will be shown by these search terms.
  • The methods selection and separation (literature values, experiment or direct measurement, expert assessment, model or GIS, stakeholder participation, etc.) need to be better explained and justified in the data descriptor.
  • Figure 1: Hard to read. No consistent layout.
  • Figure 2: Needs to be improved. Hard to understand its meaning, e.g. if you want to show a hierarchy, a classification, the CICES format, etc.?
  • The information about how the literature review (analysis) was conducted is missing, e.g. by conducting a content analysis, text mining, by using an analysis software, etc.

The most important weakness is the fact that the literature review was only conducted until January 2019 and is therefore outdated. It can be only published if the search will be expanded to January 2022. In general, the value of this data for potential users would increase if a longer time frame will be chosen, e.g. January 2012 to January 2022.

Author Response

(The authors gave the same response as above.)

Reviewer 3 Report

This manuscript can be published as a Data descriptor. Please improve figure 2, the text is not legible. Please proofread the text. Remove bold:  For example, in data scarce  regions, indicators relying on statistical or census data may not work. Under such cir1 cumstances, indicators based on field measurements or expert assessment may provide an alternative. Likewise, if time or funding are limited, experiments (and indicators based on them) may not be feasible. In this case, indicators based on literature reviews or existing statistics could be selected.

Reviewer 4 Report

Dear authors,

Your description of the dataset reflects result of a comprehensive work and have to be published. My comments are as follows:

The title of the article refers to all soil-related ecosystem services (ES) but you operated only with agricultural lands without any mention of a great area of forest, grassland and other managed and natural ecosystems.

2-       A short review on the existed ES and soil databases should be useful in the article.

3-       It is strange that there is no one ES on Soil protection.

4-       Two ESs on Soil quality (decomposition and weathering) are also strange. Soil scientists evaluate Soil quality mainly by chemical composition of soil organic matter and other physic-chemical properties and processes but these ES are absent in the dataset.

5-       Some ESs seem to be of a small value for soil-related ESs, for example Chemical composition of salt waters.

Unfortunately I failed open the Database in the Internet and check ESs more thoroughly (currently I am in Russia where German sites are not available).

I understand that all my comments could not be fixed because the structure of collected data but anyway they shall be useful to discuss them in the text.

Round 2

Reviewer 2 Report

I am grateful for the author comments. However, even though the authors invested time in the revision and tried to find arguments, it is not a scientifically convincing approach. The data base should be further expanded to more up-to-date literature (2022) and by a more comprehensive literature review by using synonyms. The screening of 156 is manageable (50 papers can be screened in one hour = 3 additional hours). And if you maybe get another 50 relevant papers to be added in the data base, it is also feasible (and valuable). The additional effort justifies the improved quality of the data set. And even though the authors say that the ecosystem services concept is fixed by the term “ecosystem services”, many scientists and practitioners who publish in scientific journals still use “ecological services”, “landscape services”, “environmental services” etc. You will miss important literature. Therefore, it is justified to expand the literature search to these terminologies. Furthermore, if I add “farm*” (synonym) in addition to “agriculture*” in your search terms in WoS, only approx. 400 entries are shown that is still manually to be handled. And if you experience large counts of literature in search engines, you can use AI-Powered Screening by e.g. by using DistillerSR. By your preselection of relevant / non-relevant, the software will facilitate your screening. Furthermore, the authors should use the asterisk as truncation: "ecosystem service*" AND indicator* in order to cover the publications where terms in plural have been used.

I am so critical since the authors indent the usage of this data set as follows: that researchers select the indicators from the data set if they have a lack of availability (access?), time or funding (line 129-130) which means that the quality of this data set will not be questioned again. If a researcher has no access or time to check the literature again, the data set should be 100% reliable. If the authors do not provide a rating or further information of suitability or indicator quality, the usage of the specific indicator can be also misleading. The scientific quality and scientific soundness of the data set should be therefore as ambitious as possible.

I think, the literature search should be expanded to improve the scientific quality that cannot be done within the short revision time given by MDPI journals. Therefore, I suggest a profound revision but would welcome a resubmission because the improved version could be of added value for the scientific community.

You can find more comments in the manuscript to improve your wording and expression.

Reviewer 4 Report

Unfortunately I have no opportunity to open sites with dadaset descrption that are mentioned on  lines  24-29 and Appendix as well because currently I am in Russia under sanction limitations. Therefore I have no access to the dataset itself. The content and style of the article's text is satisfactory.

Round 3

Reviewer 2 Report

The authors have mentioned that they did not receive the manuscript with my comments (or that they cannot read the comments). Therefore, I attached it again with another comment style. I hope that they can read my comments now.

Author Response

Thank you very much for sending the PDF version with your comments. 

Most of the comments have already been addressed in our last response and/or our last revision. We have also now taken up your suggestion to clarify that both actual and potential supply of ecosystem services are considered in our manuscript.

Round 4

Reviewer 2 Report

I am grateful that the authors have answered most of my comments. However, there are still open questions that must be answered and/ or adapted in the manuscript:

1.       Regarding the “database search was conducted in January 2019”: The justification of the fact that the literature review was only conducted until 2019 is still missing as answer to me and as text in the manuscript. Why an outdated literature review should be still published 2-3 years later? There must be a reasonable justification behind.

2.       Regarding the “search term ecosystem service in the title and indicator AND agricultur* in title, abstract or key words”: Why is the plural of “ecosystem service” automatically included if you used the word ecosystem service without asterisk? WoS or SCOPUS will not recognize the plural.

3.       Regarding “We tested the use of google scholar as a third database, rejected it due to the lower quality of search results”: First of all, it is “Google Scholar” (capital letters). Second, I kindly ask you to specify what you mean by “lower quality” and rewrite the expression in the manuscript. I think, what you mean is that it is more difficult in Google Scholar to distinguish between peer- reviewed literature and grey literature? There exist publications that are peer-reviewed but not indexed in WoS and Scopus. However, you will find them in Google Scholar. How do you deal with these cases?

4.       Regarding “ecosystem services derived directly from agricultural animals, such as the provision of food from animals, were not considered in our study”:  If you say “food from animals”, I assume that you mean meat production - that humans can use the meat after slaughter, e.g. the cow, as food? But a cow also needs fodder. Did you include fodder as ecosystem service? If not, I would like to see a justification next to line 117. And can you please rewrite the text “provision of food from animals” in order to be more clear?

5.       Regarding the different colours in Fig. 2: You can add the legend if the “different ecosystem service groups are separated by colour”. That would make it easier for the readers to distinguish between groups.

6.       You say that results are “very context specific”. It can be also context specific which indicator is suitable. However, I understand your factsheet as a non-exhaustive collection that could give some hint about possible indicators for agricultural land- and soil management. I would encourage the authors to add more text in the manuscript as critical reflection about the content that we discussed in the manuscript revision process (not only answering me personally but also making readers and potential users of the data more aware about pitfalls). I experienced that readers often take the correctness and reliability of data as granted. However, I assume that in each single document where you extracted the indicator, authors also said sth. about uncertainty and reliability of their indicator which is completely missing in your factsheet.

Looking forward to your answers. Thank you!
